

# Weak spatial-genetic structure in a native invasive, the southern pine beetle (*Dendroctonus frontalis*), across the eastern United States

Ryan C. Garrick[1], Ísis C. Arantes[1], Megan B. Stubbs[1] and Nathan P. Havill[2]

[1] Department of Biology, University of Mississippi, Oxford, MS, United States of America
[2] Northern Research Station, USDA Forest Service, Hamden, CT, United States of America

## ABSTRACT

The southern pine beetle, *Dendroctonus frontalis*, is a native pest of pine trees that has recently expanded its range into the northeastern United States. Understanding its colonization, dispersal, and connectivity will be critical for mitigating negative economic and ecological impacts in the newly invaded areas. Characterization of spatial-genetic structure can contribute to this; however, previous studies have reached different conclusions about regional population genetic structure, with one study reporting a weak east-west pattern, and the most recent reporting an absence of structure. Here we systematically assessed several explanations for the absence of spatial-genetic structure. To do this, we developed nine new microsatellite markers and combined them with an existing 24-locus data matrix for the same individuals. We then reanalyzed this full dataset alongside datasets in which certain loci were omitted with the goal of creating more favorable signal to noise ratios. We also partitioned the data based on the sex of *D. frontalis* individuals, and then employed a broad suite of genotypic clustering and isolation-by-distance (IBD) analyses. We found that neither inadequate information content in the molecular marker set, nor unfavorable signal-to-noise ratio, nor insensitivity of the analytical approaches could explain the absence of structure. Regardless of dataset composition, there was little evidence for clusters (*i.e.*, distinct geo-genetic groups) or clines (*i.e.*, gradients of increasing allele frequency differences over larger geographic distances), with one exception: significant IBD was repeatedly detected using an individual-based measure of relatedness whenever datasets included males (but not for female-only datasets). This is strongly indicative of broad-scale female-biased dispersal, which has not previously been reported for *D. frontalis*, in part owing to logistical limitations of direct approaches (e.g., capture-mark-recapture). Weak spatial-genetic structure suggests long-distance connectivity and that gene flow is high, but additional research is needed to understand range expansion and outbreak dynamics in this species using alternate approaches.

Corresponding authors
Ryan C. Garrick,
rgarrick@olemiss.edu
Nathan P. Havill,
nathan.p.havill@usda.gov

**OPEN ACCESS**

## INTRODUCTION

The southern pine beetle, *Dendroctonus frontalis* (*Zimmermann, 1868*), is an economically important pest of pine trees across eastern North America. Outbreaks have been particularly devastating to hard pines (*Gernandt et al., 2018*), and multimillion-dollar losses in timber and pulpwood have been recorded (*Price et al., 1997*; *Pye et al., 2011*). The species' impacts are now moving north, with pitch pine ecosystems in northeastern North America newly at risk (*Lesk et al., 2017*; *Dodds et al., 2018*). Given the destruction that can be caused by *D. frontalis*, and the potential for this species to continue extending its range northward owing to a warming climate, a deeper understanding of the population biology of this beetle, including development and application of molecular tools that can provide insights into dispersal, connectivity, and local mating dynamics, is of immediate importance.

Owing to the pest status of *Dendroctonus frontalis*, its natural history and population ecology are relatively well understood. All life stages are either fully or partly dependent on the inner bark or phloem of the host tree (*Hain et al., 2011*). Initially, adult pioneer female beetles are attracted to odors from stressed host trees (*e.g.*, owing to crowding, drought, disease, lightning strikes, or other storm damage), and upon arrival by flight, they initiate an attack and release aggregation pheromones that attract males (*Sullivan, 2011*). Monogamous reproduction typically occurs in nuptial chambers within a newly attacked tree, although some females arrive already gravid. Up to 30 eggs are deposited in serpentine galleries that are excavated by the female parent, who also deposits mycangial fungal spores in the galleries, leading to fungal growth that provides nutritional sustenance to the early stage larval offspring. After developing to the adult stage, offspring emerge from the outer bark of the tree, and then disperse to nearby hosts, which they attack (*Hain et al., 2011*). The combination of tree odors and aggregation pheromones can stimulate a mass attack that results in "infestation spots" that enlarge within a forest (*Sullivan, 2011*). Such large-scale irruptions cause widespread pine tree morality among healthy trees. These epidemic outbreak years have occurred at irregular intervals ranging from every 5–10 years from the early 1970s to 2000s in the species' native range in the south, where as many as nine overlapping generations per year can be completed. Historically, climatic conditions in the northern regions prevent more than just a few generations per year, with the species' latitudinal range limit thought to be constrained by winter lows, where temperatures of −16 °C cause almost 100% mortality (*Hain et al., 2011*).

There are several reasons to expect spatial-genetic structure across the range of *D. frontalis* in the eastern United States. For example, its distribution spans the southern Appalachian Mountains—a well-known biogeographic and population genetic barrier for diverse species (*Soltis et al., 2006*), including arthropods (*Garrick, 2011*). Also, given that *D. frontalis* is native to the southern United States, with a broad geographic range that extends south to Honduras and west to Arizona, there should have been ample time for such structure to evolve. Capture-mark-recapture and flight mill studies have estimated mean dispersal distances of 0.5 km to 3.4 km (*Turchin & Thoeny, 1993*; *Kinn et al., 1994*). While this is considered quite a long distance because of the species' small size (≤ three mm), given its expansion into the northeastern United States over the past decade or more

*via* source–sink stepping-stone dynamics, genetic isolation-by-distance (IBD) might be detectable. Indeed, earlier work by *Schrey et al. (2011)* identified a weak east–west division roughly coinciding with the southern Appalachian Mountains, as well as IBD across the eastern United States, using eight microsatellite loci.

Somewhat surprisingly, a recent investigation of spatial-genetic structure in *D. frontalis* by *Havill et al. (2019)* found broad-scale differentiation between eastern and western North America but did not detect distinct geo-genetic groups nor IBD within eastern North America. The discrepancy with earlier findings is not clearly attributable to differences in the underlying genetic marker set, or the extent of geographic sampling. *Havill et al. (2019)* used 24 microsatellite loci that included the eight loci from *Schrey et al. (2011)*, and the maximum pairwise distances among collection localities were very similar (1,658 km *vs.* 1,513 km, respectively). Also, in both studies, per-site sample sizes were quite large (typically ≥ 25 individuals). Thus, it would seem that biologically meaningful genetic distance estimates should have been attainable. From an applied management perspective, it is important to reconcile these contrasting inferences about spatial-genetic structure among *D. frontalis* in eastern North America because they have different ramifications for impacts upon pine forest ecosystems in newly colonized regions. For example, if distinct genetic clusters exist, multiple gene pools may converge and recombine at the wave front, producing an invasive population with elevated genetic variation and adaptive potential (*e.g.*, *Kolbe et al., 2004*). Conversely, if the species is essentially panmictic across eastern North America, then the risk of rapid adaptive evolution (*e.g.*, leading to larger population sizes and/or faster expansion speeds in novel environments; *Szücs et al., 2017*) may be lowered, and mitigation measures previously developed for outbreaks in the native range should be readily transferable to the northeastern United States. That said, forecasting responses to novel climates and associated no-analog communities is challenging (*Williams & Jackson, 2007*), likely necessitating adaptive management.

There are several plausible explanations for an unexpected absence of clusters or clines. Some can be broadly classified as technical issues, such as inadequate information content or unfavorable signal-to-noise ratio in the molecular marker set, and/or insensitivity of the chosen analytical approaches or violation of key assumptions. Alternatively, species-specific biological phenomena may mask or overwrite signatures of spatial-genetic structure. For example, when using biparentally inherited markers, a life history trait such as sex-biased dispersal could prevent detection of philopatry that is exhibited by only half of the members of a population. Likewise, a demographic event such as recent rapid range expansion could create non-equilibrium conditions that re-partition genetic variation (perhaps ephemerally) to be consistent with broad-scale panmixia (*i.e.*, similar to impacts of postglacial expansion into newly available habitats; *Hewitt, 1996*; *Hewitt, 2004*, but see *Excoffier & Ray, 2008*).

While the explanations for a lack of detectable spatial-genetic structure given above are not mutually exclusive (*e.g.*, technical limitations and biological influences may act concurrently), and acknowledging the potential for genuine panmixia (*i.e.*, a null hypothesis that may be true), the goal here was to systematically explore evidence for each of them. This was approached by augmenting, sub-setting, and reanalyzing *Havill et al.*'s (*2019*)

dataset. Specifically, we enhanced overall information content by developing and screening additional microsatellite loci, and elevated signal over noise either by identifying and omitting loci most likely to be compromised by null alleles and homoplasy, or following *Russello et al. (2012)*, by retaining only those loci that showed the greatest magnitude of differentiation among sampling sites. For each of these, we also created male-only and female-only partitions of the data. To circumvent potential analytical inefficiencies and/or restrictive assumptions, we used a suite of methods/metrics that each leverage different types of signal in the data, thereby reducing the overall risk of false negatives.

Our exploration of the extent to which inferences about the type, and strength, of spatial-genetic structure may be impacted by technical and/or biological factors improves our understanding of the biology of a destructive native invasive species. This work also contributes new molecular markers that can be used to investigate other aspects of the biology of *D. frontalis*, such as questions about parentage and relatedness, family group structure and kin clustering, as well as inbreeding dynamics.

## MATERIALS & METHODS
### General approaches to assess lack of structure
#### Inadequate information content
The power of microsatellites to detect spatial-genetic structure increases as independent loci are added (*Allendorf, Luikart & Aitken, 2013*), but in some cases, at least 30 loci may be needed to detect weak structure (*e.g.*, *Duchesne & Turgeon, 2012*). To address this, nine new microsatellite markers were developed here, and used to genotype the same individuals included in *Havill et al.*'s (*2019*) 24-locus data matrix. Briefly, those samples were collected with permission from the U.S. Department of Agriculture Forest Service by numerous colleagues (see field permit statement) between 2013 and 2017, using Lindgren funnel traps baited with frontalin and alpha-pinene. Specimens were identified as *D. fontalis* following *Armendáriz-Toledano & Zuñiga*'s (*2017*) key, and individual sex was determined based on the presence (female) or absence (male) of a mycangium on the pronotum. The new and old datasets were combined, creating what we refer to as the "augmented dataset (ADS)."

#### Unfavorable signal-to-noise ratio
The utility of microsatellite data can be compromised by technical artefacts, and/or affected (either negatively of positively) by inherent features of the loci themselves. To *reduce overall noise*, we omitted loci most likely to suffer from a higher frequency of non-amplifying null alleles, and/or allele size homoplasy based on a proxy for potential mutation rate. Herein, this is the "low noise dataset (LNDS)." As an alternative strategy, we attempted to *enhance overall signal* by retaining only those loci that maximized genetic differentiation, as measured by $F_{ST}$ (*Weir & Cockerham, 1984*) among sampling sites. The threshold for inclusion in this "high signal dataset" was chosen based on observed natural breaks in the distribution of $F_{ST}$ values.

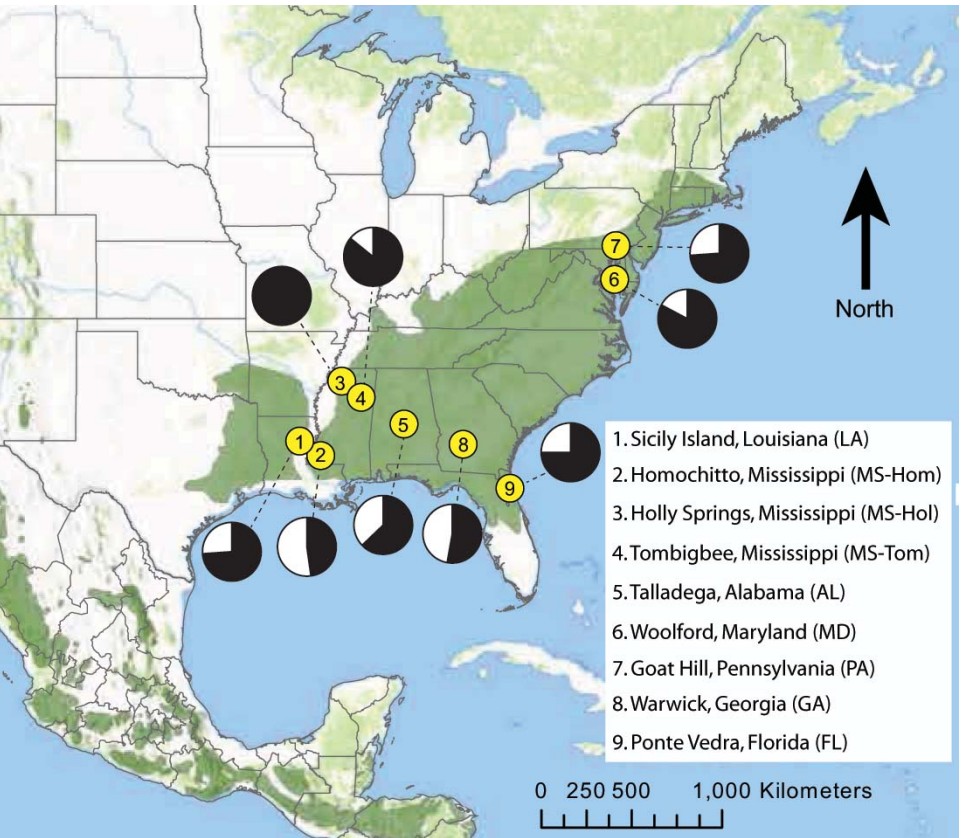

**Figure 1** **Map of eastern North America showing the locations of nine *D. frontalis* sampling sites (numbered yellow dots).** Pie charts associated with each site represent the proportion of males (black) and females (white) in the set of genotyped beetles. The geographic range of the species (green) is based on the distributions of suitable pine host tree species based on *Critchfield & Little Jr (1966)*, modified to reflect the current northern limits of *D. frontalis*.

### Sex-biased dispersal

Different degrees of site fidelity between males and females can affect detection of population differentiation when using biparentally inherited microsatellite markers. Impacts are most pronounced when sex-specific dispersal rates and distances are highly asymmetric, and when sampling includes many individuals that have dispersed but not yet reproduced (*Prugnolle & De Meeûs, 2002*). To evaluate this, we analyzed male-only ($n = 185$ individuals from 9 sites) and female-only ($n = 70$, 8 sites) partitions, in addition to combined datasets ($n = 255$, 9 sites; Fig. 1).

### Insensitivity of analysis, or violation of assumptions

Genotypic clustering methods differ in their incorporation of geo-spatial information, and when differentiation is weak, this impacts accuracy (*Chen et al., 2007*). Also, sensitivity of IBD analyses can depend on the chosen genetic distance metric (*Séré et al., 2017*). To at least partly account for this, each of the above datasets were analyzed using several clustering and IBD approaches that represent different strengths and limitations of a broader suite

of existing methods and metrics. That said, we recognize that clusters *vs.* clines are not mutually exclusive (*Rosenberg et al., 2005*), and so we interpret these outcomes with caution, and in the context of our sampling limitations.

## Development and validation of new microsatellite loci

In the present study, we extended upon the assessment of a suite of microsatellite loci identified by *Havill et al. (2019)*. Specifically, nine additional loci were selected from among those identified using the software QDD (*Meglecz et al., 2014*) following low coverage whole genome sequencing of one male *D. frontalis* from Homochitto National Forest, Mississippi (NCBI BioProject Accession Number PRJNA493650). To assess the suitability of these new loci, multiplex polymerase chain reaction (PCR) amplifications were performed for sets of three loci, each with a different 5′ fluorescent label on the forward primer and mostly non-overlapping allele size ranges (Table 1). Reactions were conducted in 15 μL volumes that contained 7.5 μL Type-it® Microsatellite PCR Kit master mix (Qiagen, Valencia CA), 1.5 μL dH$_2$O, 0.75 μL of each primer (10 mM), and 1.5 μL genomic DNA. The following "touchdown" thermal cycling conditions were used: 95 °C for 2 min (1 cycle), 95 °C for 30 s, 61 °C minus 2 °C per successive cycle for 30 s, 72 °C for 45 s (5 cycles), 95 °C for 30 s, 51 °C for 30 s, 72 °C for 45 s (30 cycles), and 60 °C for 30 min (1 cycle).

Amplified fragments were run on an ABI 3730 sequencer (Life Technologies) with a Liz 500 ladder (Gel Company, San Francisco CA) at Yale University's DNA Analysis Facility on Science Hill. Genotype scoring was performed using GENEIOUS v10.0.5 (*Kearse et al., 2012*). Following *Havill et al. (2019)*, we assessed Mendelian inheritance patterns for each of the new loci in two local populations (*i.e.*, 27 individuals from Sicily Island, Louisiana, and 28 individuals from Tombigbee, Mississippi). For each locus and population, evidence for departures from Hardy-Weinberg Equilibrium (HWE) were examined *via* exact tests (*Guo & Thompson, 1992*), with significance assessed using 10,000 Markov Chain permutations. Observed ($H_O$) and expected ($H_E$) heterozygosity were also calculated. Independent segregation of alleles among loci was assessed using all *D. frontalis* samples (255 individuals) by testing for departures from Linkage Equilibrium (LE) for all possible pairs of 33 loci (*i.e.*, nine reported here, plus 24 from *Havill et al., 2019*), again using 10,000 Markov Chain permutations. Given that nine different sampling sites were included, Fisher's combined probability test was used to calculate *P*-values across populations for each locus pair. These analyses were performed in GENEPOP v4.5.1 (*Rousset, 2008*), with sequential Bonferroni correction (*Holm, 1979*) of *P*-values associated with LE tests.

## Identification and omission of loci that may elevate stochastic noise

MICRO-CHECKER v2.2.3 (*Van Oosterhout et al., 2004*) was used to determine whether there was evidence for null alleles, and if so, to estimate their frequencies. For each locus in each local population, a null allele was inferred to exist if there was a significant excess of homozygotes that were evenly distributed across all homozygote classes. This was determined by Fisher's combined probability test, with *P*-values determined using 1,000 permutations. Frequency of the null allele ($r$) was then was estimated using the method of *Chakraborty et al. (1992)*, which assumes that missing data are not attributable

Garrick et al. (2021), *PeerJ*, DOI 10.7717/peerj.11947

**Table 1** **Characterization of microsatellite loci for *D. frontalis* from each of two local populations.** As in *Havill et al. (2019)*, a pig-tail (5′-GTTT-3′) was added to the 5′ end of each reverse primer (shown below).

| Locus name | Primer sequences 5′ to 3′ (fluorescent label) | Repeat motif | All ENA populations | | Sicily Island, Louisiana | | | Tombigbee, Mississippi | | |
|---|---|---|---|---|---|---|---|---|---|---|
| | | | Allele size range (bp) | No. of alleles | $H_O$ | $H_E$ | No. of alleles | $H_O$ | $H_E$ | No. of alleles |
| SPB4422 | F: (FAM)-ATCGACTTCGCACGCAAAAC<br>R: GTTTCCGCTTTCACTCACTTTAATCAT | AC | 226–246 | 7 | 0.259 | 0.236 | 5 | 0.179 | 0.165 | 3 |
| SPB903595 | F: (NED)-TTTATGTCTATGCCGGATGG<br>R: GTTTGGACATTGACAAAATCGGAC | CAG | 262–278 | 7 | 0.320 | 0.269 | 2 | 0.250 | 0.433 | 3 |
| SPB180144 | F: (PET)-ACTAATATTTCAGGTCCGCC<br>R: GTTTGAGCTACTGAAAATTGCGAC | AC | 172–182 | 6 | 0.185[*] | 0.294 | 4 | 0.429 | 0.443 | 4 |
| SPB265317 | F: (FAM)-AAACATGTCGGGGAATCTAC<br>R: GTTTGTTCATTAGCAGCAGGGATA | AT | 376–395 | 10 | 0.593 | 0.707 | 5 | 0.593 | 0.713 | 5 |
| SPB979494 | F: (NED)-TGACATATGCGACATAAGGG<br>R: GTTTGAAGTGTTTATTGTGCTCGG | ATC | 192–225 | 8 | 0.630 | 0.502 | 4 | 0.679 | 0.561 | 5 |
| SPB4155 | F: (PET)-GATGCAGTGAAAGTGGCGTG<br>R: GTTTGCCGATCTTTACCAACTCAAGC | ATG | 95–116 | 8 | 0.481 | 0.519 | 4 | 0.593 | 0.562 | 4 |
| SPB3702 | F: (FAM)-AACGCTTCACATTTGCACCG<br>R: GTTTCATCGGATAATCCTGCGGGA | CAC | 93–117 | 10 | 0.185 | 0.173 | 4 | 0.321 | 0.284 | 4 |
| SPB1278 | F: (VIC)-TCAGATCTGAGACGACAAGAAAGA<br>R: GTTTCCGGTCTGCAAATACGAGGT | AT | 104–120 | 7 | 0.259 | 0.233 | 4 | 0.357[*] | 0.390 | 5 |
| SPB1534 | F: (PET)-CGGGTGAAAGAGTTAGGGGA<br>R: GTTTGCCCTTACGATCACAGGTACT | (CCA)<br>GAA<br>(CCA) | 109–133 | 10 | 0.615[*] | 0.641 | 6 | 0.643 | 0.645 | 6 |

**Notes.**

Abbreviations: ENA, Eastern North America; ($H_O$), observed heterozygosity; $H_E$, expected heterozygosity; F, forward primers; R, reverse primers.

*$H_O$ values with an asterisk denote loci that showed significant departure from Hardy-Weinberg Equilibrium at the 0.05-level.

to homozygous nulls, and that populations are panmictic. To identify inbred populations for removal, we relied on the tendency for inbreeding to generate homozygote excess at many loci (cf. locus-specific impacts of null alleles). Here we defined outbred populations as those in which fewer than seven loci (*i.e.,* <20%) showed significantly ($P < 0.05$) positive $F_{IS}$ values. For each locus and population, the null hypothesis of panmixia was compared to an alternative hypothesis of heterozygote deficiency using a U-test (*Rousset & Raymond, 1995*), performed in GENEPOP with 10,000 Markov Chain permutations. After omitting outbred populations, we rank-ordered the loci based on the level of noise contributed by putative null alleles. To ensure that both prevalence and severity of the null allele at a locus were considered, the *r* value from each population with a significant excess of homozygotes was summed ($r_{cumulative}$).

The number of contiguous repeat units may be a useful proxy for microsatellite mutation rate, given that DNA structure affects the opportunity for slipped-strand mispairing (*Bhargava & Fuentes 2010*, and references therein). Here we used the reported positive association between repeat number and mutation rate to identify which loci might be predisposed to having alleles identical in state but not by descent. Briefly, based on one sequence per locus generated during marker development (*Schrey et al., 2007*; *Havill et al., 2019*; this study), we first estimated the PCR amplicon length for each locus given the locations of forward and reverse primer annealing sites. Next, the observed number of contiguous repeats was scaled to reflect what should be present within the median-sized allele (*i.e.,* a reference allele identified from the eastern United States gene pool as whole). Rank-ordering of loci based on the potential level of stochastic noise contributed by homoplasy followed the inferred number of repeats of each reference allele.

The omission of loci that may elevate overall noise was based on joint consideration of null alleles and homoplasy. Along each of these two axes, we iteratively omitted the lowest ranked locus one at a time, until a two-sample *t*-test (one-tailed) determined that a significant ($P < 0.05$) reduction in the mean value of the original metric used for rank-ordering (compared to that of the 33-locus dataset) had been achieved. To assess whether the low noise dataset was comprised of a subset of loci with inadvertently low information content, we calculated *Paetkau et al.*'s (*1995*) Probability of Identity (PI; the chance that two individuals randomly drawn from a panmictic population have the same genotype) using GENALEX v6.5 (*Peakall & Smouse, 2006*; *Peakall & Smouse, 2012*). We then compared mean PI between the reduced *vs.* full dataset, using a *t*-test as above.

## Identification and retention of loci that may enhance overall signal

For species in which population divergence is shallow, "outlier" microsatellite loci (*e.g.,* those displaying unusually strong differentiation, potentially due to effects of divergent selection on linked genomic regions) may be particularly useful for identification of population structure (*Russello et al., 2012*). This is because recent divergence (*e.g.,* over post-glacial timescales or shorter) may be slow to register in most loci, particularly when it is coupled with large effective populations sizes. To explore the potential benefit of focusing on loci with the strongest signatures of spatial-genetic structure in *D. frontalis*, we calculated global $F_{ST}$ (*Weir & Cockerham, 1984*) using the augmented dataset, for each

of the 33 loci separately, and then plotted the distribution of these values in rank order. Natural breaks were identified qualitatively, based on the magnitude of change (*i.e.,* $\Delta F_{ST}$ $= F_{\text{ST locus } n} - F_{\text{ST locus } n+1}$, where $n$ is the rank order position). These breaks defined the thresholds for retention *vs.* exclusion of loci from the high signal dataset.

## Genotypic clustering analyses

To infer the number of natural genetic groups ($K$) and their members, we analyzed microsatellite datasets using three approaches: STRUCTURE v2.3.4 (*Pritchard, Stephens & Donnelly, 2000*), BAPS v6.0 (*Corander, Waldmann & Sillanpää, 2003*), and Discriminant Analysis of Principal Components (DAPC; *Jombart, Devillard & Balloux, 2010*). The first two are Bayesian approaches and assume HWE and LE within "true" clusters. However, STRUCTURE is strictly individual-based and does not make use of geo-referenced samples (although priors on population of origin can be applied), but it has been shown to tolerate clinal variation relatively well (*Chen et al., 2007*). BAPS can be run using either individuals or groups of individuals as the basic units of analysis, and it explicitly considers spatial coordinates, which can help when differentiation is weak. DAPC can also be used to identify natural groups of individuals *a posteriori*, but this method does not attribute the underlying cause of such groups to population genetic processes, nor does it incorporate spatial information. In all clustering analyses, we assessed $K$ values from one up to and including the total number of sampling sites (*i.e.,* maximum $K = 8$ or 9, depending on the dataset under consideration).

*STRUCTURE.* All runs employed the correlated allele frequency and admixture ancestry models. Estimated -ln likelihood scores were obtained for each value of $K$, with 20 replicates each. A burn-in of 50,000 Markov chain Monte Carlo (MCMC) generations and run length of 200,000 MCMC generations were used, with other parameters set as default. The best-fit value of $K$ was chosen using *Evanno, Regnaut & Goudet*'s (*2005*) $\Delta K$ method implemented in STRUCTURE HARVESTER v0.6.94 (*Earl & vonHoldt, 2012*). However, the $\Delta K$ method is unable to assess support for $K = 1$, and it may be subject to bias with respect to identifying only the top (*e.g.,* $K = 2$) level of hierarchical structure (see *Janes et al., 2017*). Accordingly, we also used a visual assessment of Ln Pr(X|$K$) plots, from which the smallest $K$ that captured the major structure in the data was accepted, following *Pritchard & Wen (2003)*.

*BAPS.* To enable spatial clustering of individuals, we created non-redundant GPS coordinates *via* the addition of fine-scale jitter (*i.e.,* a random number at the 4th and 5th decimal places, equivalent to $\leq 11.1$ m at the equator). For spatial clustering of groups of individuals, each of the 8 or 9 sampling locations (depending on the dataset) defined an *a priori* group, and the original spatial coordinates were used. Accordingly, the complexity of Voronoi tessellations differed considerably between the two approaches. Spatial data were analyzed with associated microsatellite genotype data, with 10 replicates per run. The best-fit $K$ was identified *via* log marginal likelihood scores (*Cheng et al., 2013*).

*DAPC.* Individual-based multilocus microsatellite genotypes were first transformed *via* principal component (PC) analysis to identify uncorrelated variables to be used for subsequent discriminant analysis. The best-fit number of clusters was determined *via* $k$-means clustering, with the lowest Bayesian Information Criterion (BIC) score used

to choose *K*. Following *Miller, Cullingham & Peery (2020)*, an initial preliminary DAPC was conducted to determine the optimal number of PCs to retain, followed by a second final DAPC using the chosen number of PCs. In the initial assessment, the optimal number of PCs was determined using the cross-validation procedure, considering 20 to 300 PCs (in increments of 20) with 30 replicates at each level of PC retention, and 90% of the data comprising the training set. Pre-defined groups based on sampling location information were used only for the purpose of quantifying the number of *de novo* clusters that were inferred to be present at a given sampling location. DAPC was implemented using ADEGENET v2.1.2 (*Jombart, 2008*) in R v3.6.1 (*R Core Team, 2020*).

## Isolation-by-distance analyses

To assess evidence for subtle gradients in genetic variation across the landscape, we conducted IBD analyses of our microsatellite datasets using either sampling sites or individuals as the basic units. The rationale for these two approaches was that population allele frequencies typically can change rather gradually over time and therefore provide insights into historical processes, whereas diploid genotypes are much more labile owing to reshuffling of alleles every generation in sexually reproducing species, and therefore have the potential to closely track recent or on-going processes (*Sunnucks, 2000*; *Garrick, Caccone & Sunnucks, 2010*; *Epps & Keyghobadi, 2015*).

*Population-based metrics.* Three measures of genetic distance were used: *Cavalli-Sforza & Edwards*'s (*1967*) chord distance ($D_c$), which has desirable properties for detecting IBD in the presence of null alleles (*Séré et al., 2017*); *Nei*'s (*1972*) standard genetic distance ($D_s$), which increases linearly with geographic distance if mutation follows an infinite allele model and occurs at a constant rate; and *Weir & Cockerham*'s (*1984*) estimate of $F_{ST}$ linearized as $F_{ST}/(1-F_{ST})$ following *Slatkin (1995)*, to account for the high mutation rate of microsatellite loci.

*Individual-based metrics.* Three metrics for genetic similarity were used: the kinship coefficient ($D_{kf}$, *Cavalli-Sforza & Bodmer, 1971*); proportion of shared alleles ($D_{ps}$, *Bowcock et al., 1994*), or relatedness ($\hat{r}$, *Lynch & Ritland, 1999*). Briefly, for a given locus, $D_{kf}$ is the probability that a randomly chosen allele from one individual is identical-by-descent to that of second individual (summed over all loci and alleles). Conversely, when comparing two mutlilocus genotypes, $D_{ps}$ is calculated as number of shared alleles summed over loci / ($2\times$ number of loci compared). Unlike the other two metrics, *Lynch & Ritland*'s (*1999*) methods-of-moments estimator of relatedness makes use of information on population allele frequencies (*e.g.*, a shared rare allele is considered more likely to be identical-by-descent than a shared common allele). Given that $\hat{r}$ is asymmetrical, following *Ritland (2000)*, we used the average of both directions.

MICROSATELLITE ANALYZER v4.05 (*Dieringer & Schlötterer, 2003*) was used to calculate $D_c$, $D_{kf}$, and $D_{ps}$, whereas GENALEX was used to calculate $D_s$, $F_{ST}$ and $\hat{r}$, as well as linear pairwise geographic distance matrices based on latitude and longitude coordinates, using a modified Haversine Formula. For all datasets and distance metrics, the significance of IBD was assessed *via* *Mantel (1967)* tests with 999 permutations.

## RESULTS

### Development and validation of new microsatellite loci

For the nine new loci developed here, amplification success across all eastern United States *D. frontalis* samples was reasonably high (mean of 4.2% missing data per locus). Three loci showed significant departures from HWE in the direction of homozygote excess. However, in all cases this was limited to just one of the two local collection sites (Table 1), suggesting that there were no intrinsic issues with allelic inheritance (*i.e.,* the new microsatellite loci appear to be autosomal, diploid, and single copy). Levels of with-population polymorphism were modest (mean of ∼4.3 alleles per new locus for the two exemplar local populations). Across all eastern United States *D. frontalis* samples, there was a mean of 8.1 alleles per new locus (Table 1). Based on our assessment of LE for all 33 loci (528 locus pairs), 22 pairs were significant at the $P <0.05$-level, but none remained so after sequential Bonferroni correction to account for multiple tests. Accordingly, there was no strong evidence for non-independence among loci in the augmented dataset. Preliminary analyses using all loci indicated weak genetic structure among sampling sites ($F_{ST} = 0.009$, $P = 0.001$).

### Identification and omission of loci that may elevate overall noise

*Null alleles.* Two local populations had seven or more loci with significantly positive $F_{IS}$ values: Homochitto, MS, and Woolford, MD. These were considered inbred and therefore omitted from consideration when assessing null alleles. MICRO-CHECKER identified 13 loci that may have a null allele (Table 2). However, prevalence was low (*i.e.,* restricted to a single population) for 10 of these. Of the remaining loci, two had null alleles in four populations, and one had nulls in two populations. Across the 20 cases of putative null alleles (out of 33 loci ×7 retained populations = 231 potential cases), mean estimated $r$ was 0.245 (range: 0.139–0.431). Rank-ordering followed by iterative removal revealed that omission of the five worst loci was required to significantly reduce the mean $r_{cumulative}$ value ($t = 1.720$, d.f. $= 40$, $P = 0.047$). The removal of these loci did not cause a concomitant reduction in mean information content, as measured by PI ($t = -0.435$, d.f. $= 57$, $P = 0.332$).

*Homoplasy.* For 31 loci, the available allele sequence trimmed at the 5′ ends of both primers yielded a predicted PCR product that was within the empirically determined size range for eastern United States *D. frontalis.* For two loci with unexpectedly short or long amplicons, an alternative priming site (*i.e.,* a region of high sequence similarity located up or downstream of the original target site) was readily identifiable (Table 3). Three loci constrained two distinct microsatellite regions. These were either composed of different (*i.e.,* compound) repeat motifs located immediately adjacent to one other (locus Dfr-16; *Schrey et al., 2008*), or the same (*i.e.,* impure) repeat motif interrupted by a 2-bp (locus Dfr-24; *Schrey et al., 2008*) or 3-bp (locus SPB1534; this study) insertion. In all three cases, only the longest repeat region was used as a proxy for potential for homoplasy. Following re-scaling (see Methods), across 33 loci the average value of the median contiguous number of repeats was 7.6 (range: 4.945–14.901; Table 3). The five most concerning loci needed to be removed in order to significantly reduce the number of repeat units ($t = 1.819$, d.f.

**Table 2 MICRO-CHECKER assessment of null alleles and estimates of their frequency ($r$).** Twenty cases of putative null alleles were detected (marked by an asterisk) across 33 loci and seven outbred populations (abbreviations follow Fig. 1). For each locus, $r_{cumulative}$ is the sum of $r$ values from each population with a significant excess of homozygotes, and this was used to rank-order loci from worst (1st) to best (equal 14th).

| Locus name | Estimated null allele frequency ($r$) per population | | | | | | | $r_{cumulative}$ | Locus rank |
|---|---|---|---|---|---|---|---|---|---|
| | AL | FL | GA | LA | MS-Hol | MS-Tom | PA | | |
| SPB2727 | 0.000 | 0.000 | 0.000 | 0.116 | 0.273 | 0.000 | 0.000 | 0.000 | 14 |
| Dfr-09 | 0.000 | 0.043 | 0.000 | 0.021 | 0.000 | 0.071 | 0.000 | 0.000 | 14 |
| Dfr-16 | 0.001 | 0.000 | 0.096 | 0.000 | 0.008 | 0.000 | 0.000 | 0.000 | 14 |
| SPB3731 | 0.000 | 0.000 | 0.140 | 0.008 | 0.000 | 0.104 | 0.000 | 0.000 | 14 |
| Dfr-10 | 0.055 | 0.000 | 0.000 | 0.029 | 0.000 | 0.011 | 0.000 | 0.000 | 14 |
| SPB2313 | 0.080 | 0.046 | 0.033 | 0.000 | 0.156 | 0.005 | 0.002 | 0.000 | 14 |
| SPB0138 | 0.000 | 0.000 | 0.000 | 0.000 | 0.000 | 0.000 | 0.000 | 0.000 | 14 |
| SPB1983 | 0.055 | 0.000 | 0.015 | 0.000 | 0.000 | 0.000 | 0.000 | 0.000 | 14 |
| Dfr-24 | 0.179* | 0.043 | 0.000 | 0.016 | 0.139* | 0.000 | 0.011 | 0.318 | 5 |
| Dfr-14 | 0.093 | 0.000 | 0.000 | 0.000 | 0.000 | 0.082 | 0.000 | 0.000 | 14 |
| Dfr-17 | 0.010 | 0.013 | 0.082 | 0.000 | 0.000 | 0.050 | 0.000 | 0.000 | 14 |
| SPB2613 | 0.033 | 0.327* | 0.021 | 0.000 | 0.000 | 0.012 | 0.000 | 0.327 | 4 |
| SPB1230 | 0.020 | 0.000 | 0.000 | 0.000 | 0.000 | 0.000 | 0.000 | 0.000 | 14 |
| Dfr-18 | 0.032 | 0.000 | 0.024 | 0.000 | 0.021 | 0.000 | 0.000 | 0.000 | 14 |
| SPB1875 | 0.000 | 0.009 | 0.000 | 0.000 | 0.000 | 0.000 | 0.000 | 0.000 | 14 |
| SPB3013 | 0.042 | 0.000 | 0.000 | 0.000 | 0.000 | 0.000 | 0.136 | 0.000 | 14 |
| SPB1272 | 0.017 | 0.000 | 0.000 | 0.000 | 0.000 | 0.000 | 0.043 | 0.000 | 14 |
| SPB1284 | 0.000 | 0.041 | 0.000 | 0.000 | 0.248* | 0.052 | 0.013 | 0.248 | 7 |
| SPB1242 | 0.044 | 0.000 | 0.000 | 0.044 | 0.000 | 0.142 | 0.000 | 0.000 | 14 |
| SPB2480 | 0.431* | 0.134 | 0.000 | 0.208 | 0.324* | 0.246* | 0.404* | 1.405 | 1 |
| Dfr-06 | 0.048 | 0.044 | 0.046 | 0.054 | 0.069 | 0.099 | 0.054 | 0.000 | 14 |
| SPB1507 | 0.162* | 0.089 | 0.235* | 0.085 | 0.197* | 0.094 | 0.249* | 0.843 | 2 |
| SPB2187 | 0.014 | 0.000 | 0.071 | 0.058 | 0.000 | 0.148* | 0.064 | 0.148 | 12 |
| SPB1270 | 0.161* | 0.106 | 0.000 | 0.000 | 0.156 | 0.000 | 0.000 | 0.161 | 10 |
| SPB4422 | 0.000 | 0.000 | 0.000 | 0.000 | 0.000 | 0.000 | 0.239* | 0.239 | 8 |
| SPB903595 | 0.000 | 0.000 | 0.106 | 0.000 | 0.053 | 0.268* | 0.000 | 0.268 | 6 |
| SPB180144 | 0.024 | 0.025 | 0.000 | 0.228* | 0.083 | 0.016 | 0.012 | 0.228 | 9 |
| SPB265317 | 0.054 | 0.094 | 0.149* | 0.088 | 0.000 | 0.092 | 0.083 | 0.149 | 11 |
| SPB979494 | 0.000 | 0.000 | 0.078 | 0.000 | 0.000 | 0.000 | 0.068 | 0.000 | 14 |
| SPB4155 | 0.053 | 0.000 | 0.426* | 0.037 | 0.076 | 0.000 | 0.000 | 0.426 | 3 |
| SPB3702 | 0.006 | 0.006 | 0.000 | 0.000 | 0.065 | 0.000 | 0.223 | 0.000 | 14 |
| SPB1278 | 0.000 | 0.000 | 0.000 | 0.000 | 0.073 | 0.044 | 0.000 | 0.000 | 14 |
| SPB1534 | 0.145* | 0.000 | 0.000 | 0.021 | 0.026 | 0.002 | 0.069 | 0.145 | 13 |

$= 50$, $P = 0.037$), but their omission did not also reduce mean PI ($t = -0.780$, d.f. $= 58$, $P = 0.219$).

Of the two sets of five omitted loci, one (Dfr-24) was common to both approaches used to reduce noise. Accordingly, the largest subset of loci with reduced null allele and homoplasy issues was made up of 24 loci. Reassessment of the impacts of our strategic removal of nine loci to create the low noise dataset showed that the reduction in mean

**Table 3  Characterization of a proxy for microsatellite mutation per locus.** From each directly sequenced allele, the number of contiguous repeat units was recorded from the longest uninterrupted microsatellite region (imperfect or compound repeats were seen in three loci, marked with †), and allele length was determined by trimming sequences at the ends of primer annealing sites (alternative priming sites were inferred for two loci, marked with #). For each locus, all allele sizes observed across Eastern North American (ENA) populations ($n = 255$ individuals) were used to calculate the median allele size, and the associated number repeat units was extrapolated and then used to rank-order loci from most (1st) to least (33rd) potential for homoplasy.

| Locus name | One directly sequenced allele | | | ENA population screening | | Locus rank |
|---|---|---|---|---|---|---|
| | NCBI accession | No. of contiguous repeat units | Allele length (bp) | Median allele length (bp) | Median no. of contiguous repeat units | |
| SPB2727 | PRJNA493650 | 7 | 144 | 141.5 | 6.88 | 19 |
| Dfr-09 | EF126297 | 10 | 109 | 104.5 | 9.59 | 5 |
| Dfr-16 | EF126300 | 11 † | 191 | 183 | 10.54 | 3 |
| SPB3731 | PRJNA493650 | 6 | 145 | 142 | 5.88 | 31 |
| Dfr-10 | EF126298 | 13 | 158 | 149.5 | 12.30 | 2 |
| SPB2313 | PRJNA493650 | 9 | 296 | 292.5 | 8.89 | 9 |
| SPB0138 | PRJNA493650 | 6 | 115 # | 116 | 6.05 | 27 |
| SPB1983 | PRJNA493650 | 7 | 143 | 142 | 6.95 | 18 |
| Dfr-24 | EF126305 | 15 † | 151 | 150 | 14.90 | 1 |
| Dfr-14 | EF126299 | 9 | 188 | 190 | 9.10 | 8 |
| Dfr-17 | EF126301 | 9 | 122 | 126 | 9.30 | 6 |
| SPB2613 | PRJNA493650 | 7 | 283 | 291.5 | 7.21 | 14 |
| SPB1230 | PRJNA493650 | 7 | 219 | 227.5 | 7.27 | 13 |
| Dfr-18 | EF126302 | 7 | 130 | 136.5 | 7.35 | 12 |
| SPB1875 | PRJNA493650 | 6 | 110 | 118 | 6.44 | 22 |
| SPB3013 | PRJNA493650 | 6 | 154 | 156.5 | 6.10 | 25 |
| SPB1272 | PRJNA493650 | 9 | 263 | 259.5 | 8.88 | 10 |
| SPB1284 | PRJNA493650 | 8 | 130 | 126 | 7.75 | 11 |
| SPB1242 | PRJNA493650 | 6 | 146 | 144 | 5.92 | 30 |
| SPB2480 | PRJNA493650 | 6 | 162 | 164.5 | 6.09 | 26 |
| Dfr-06 | EF126295 | 10 | 139 | 139 | 10.00 | 4 |
| SPB1507 | PRJNA493650 | 7 | 146 | 149 | 7.14 | 15 |
| SPB2187 | PRJNA493650 | 6 | 196 | 194 | 5.94 | 29 |
| SPB1270 | PRJNA493650 | 9 | 243 | 246 | 9.11 | 7 |
| SPB4422 | PRJNA493650 | 7 | 244 # | 230 | 6.60 | 21 |
| SPB903595 | PRJNA493650 | 5 | 275 | 272 | 4.95 | 33 |
| SPB180144 | PRJNA493650 | 7 | 174 | 177 | 7.12 | 16 |
| SPB265317 | PRJNA493650 | 6 | 386 | 389 | 6.05 | 28 |
| SPB979494 | PRJNA493650 | 7 | 215 | 205.5 | 6.69 | 20 |
| SPB4155 | PRJNA493650 | 7 | 107 | 108 | 7.07 | 17 |
| SPB3702 | PRJNA493650 | 6 | 109 | 103.5 | 5.70 | 32 |
| SPB1278 | PRJNA493650 | 6 | 108 | 110 | 6.11 | 24 |
| SPB1534 | PRJNA493650 | 6 † | 116 | 120.5 | 6.23 | 23 |

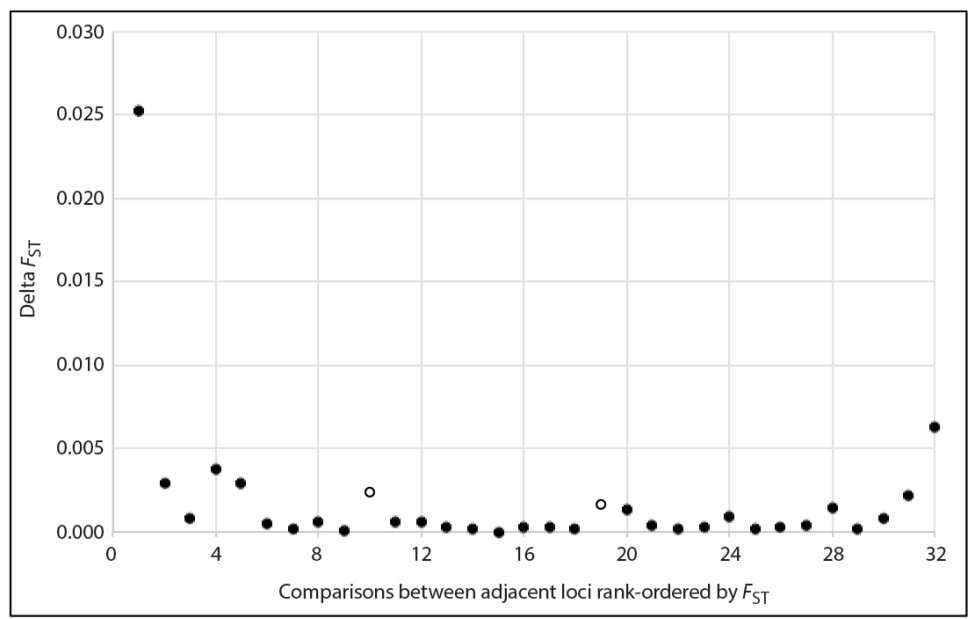

**Figure 2** Plot showing the magnitude of reduction in $F_{ST}$ between adjacent pairs of rank-ordered loci, used to visually identify inflection points (two open circles) to guide choice of thresholds for the number loci included in the high sensitivity datasets.

$r_{\text{cumulative}}$ remained close to significant ($t = 1.520$, d.f. $= 42$, $P = 0.068$), and so did reduction in mean number of repeats ($t = 1.499$, d.f. $= 53$, $P = 0.070$). As before, there was no inadvertent change in mean PI ($t = -1.180$, d.f. $= 51$, $P = 0.122$). Although our low noise dataset happened to contain the same number of loci originally analyzed by *Havill et al. (2019)*, their compositions differed by eight loci.

## Identification and retention of loci that may enhance overall signal

Based on a visual examination of the $\Delta F_{ST}$ plot, several inflection points marking pronounced decreases in $F_{ST}$ among adjacent rank-ordered loci were apparent (Fig. 2). Aside from steep declines that would have set a threshold for retention of too few loci for meaningful population genetic analyses (*i.e.*, $\leq 5$ loci), there were two other inflection points: one at the transition from the 10th to 11th ranked locus, and another at the transition from 19th to 20th. Accordingly, we created two alternative versions of the high sensitivity dataset (HSDS). One contained 10 loci with the highest $F_{ST}$ values, and the other was an expansion of this, where we retained the 19 highest ranked loci (Table 4). The 10-locus and 19-locus high signal datasets contained 3 and 6 loci, respectively, that were not shared with the 24-locus low noise dataset. Compared to *Havill et al.*'s (*2019*) original dataset, the 10-locus and 19-locus high signal datasets contained 2 and 5 novel loci, respectively. Ultimately, all datasets analyzed in the present paper differed from each other and from the previous study, thereby enabling meaningful comparisons.

**Table 4** Global $F_{ST}$ values for each microsatellite locus, calculated across nine sampling sites ($n = 255$ individuals), and their rank-ordering from highest (1st) to lowest (33rd).

| Locus name | Global $F_{ST}$ | Locus rank |
|---|---|---|
| SPB2727 | 0.025 | 11 |
| Dfr-09 | 0.027 | 9 |
| Dfr-16 | 0.023 | 15 |
| SPB3731 | 0.035 | 4 |
| Dfr-10 | 0.022 | 18 |
| SPB2313 | 0.014 | 31 |
| SPB0138 | 0.006 | 33 |
| SPB1983 | 0.018 | 25 |
| Dfr-24 | 0.024 | 12 |
| Dfr-14 | 0.019 | 22 |
| Dfr-17 | 0.028 | 8 |
| SPB2613 | 0.018 | 24 |
| SPB1230 | 0.012 | 32 |
| Dfr-18 | 0.017 | 27 |
| SPB1875 | 0.022 | 19 |
| SPB3013 | 0.028 | 7 |
| SPB1272 | 0.027 | 10 |
| SPB1284 | 0.028 | 6 |
| SPB1242 | 0.019 | 23 |
| SPB2480 | 0.064 | 1 |
| Dfr-06 | 0.015 | 29 |
| SPB1507 | 0.017 | 26 |
| SPB2187 | 0.036 | 3 |
| SPB1270 | 0.023 | 16 |
| SPB4422 | 0.023 | 14 |
| SPB903595 | 0.031 | 5 |
| SPB180144 | 0.017 | 28 |
| SPB265317 | 0.023 | 17 |
| SPB979494 | 0.023 | 13 |
| SPB4155 | 0.039 | 2 |
| SPB3702 | 0.021 | 20 |
| SPB1278 | 0.015 | 30 |
| SPB1534 | 0.019 | 21 |

## Genotypic clustering analyses

*STRUCTURE.* Across all 12 datasets (*i.e.,* ADS, LNDS, and HSDS with 10 or 19 loci; males and females combined, or separated) there were five instances where the two alternative approaches for identifying the best-fit value of $K$ were in conflict. In these cases, *Pritchard & Wen*'s (*2003*) qualitative method inferred $K = 1$ whereas *Evanno, Regnaut & Goudet*'s (*2005*) $\Delta K$ method inferred $K = 2$, 3, or 4, depending in the dataset (Table 5; but note that $\Delta K$ is not capable of assessing $K = 1$). More generally, irrespective of whether there was agreement or conflict between methods, wherever $K > 1$ was inferred, the optimal clustering

solution almost always contained "ghost clusters" (*sensu Guillot et al., 2005*); *i.e.,* those for which no individuals were strongly assigned with a membership coefficient of $Q>0.50$). In some cases, all clusters were nonsensical. For instance, based on $\Delta K$, the best-fit value was $K=3$ for LNDS males plus females, and LNDS females-only, yet not a single individual was strongly assigned; Table 5). Additionally, when $K>1$ there was also an absence of geographic localization of clusters. Ultimately, as with *Havill et al.*'s (*2019*) STRUCTURE analysis of a 24-locus *D. frontalis* dataset, our reanalysis using augmented and/or sub-setted datasets did not provide any indications of discrete population structure.

*BAPS.* The optimal partition of geo-referenced molecular data was $K=1$ for all datasets, using both individual- and population-based clustering approaches (Table 5). Given this outcome, we performed post-hoc HWE exact tests in GENEPOP, treating all individuals as members of a single population. Each of the 12 datasets showed highly significant departures from HWE (all $P<0.0005$), indicating that eastern United States *D. frontalis* cannot simply be characterized as panmictic.

*DAPC.* Across all datasets, there were only two instances where $K=1$ was inferred (Table 5; but note that DAPC was not intended for this purpose; see *Miller, Cullingham & Peery, 2020*). However, wherever $K\geq2$, there were strong indications that the groups were artefacts rather true reflections of population structure. For example, there were no cases where a given local sampling site contained members of only a single inferred cluster. Indeed, in seven of the datasets with best-fit $K\geq2$, all sampling sites contained representatives of all inferred clusters (Table 5); such levels of coexistence among putatively distinct gene pools has no plausible biological explanation.

## Isolation-by-distance analyses

*Population-based metrics.* Regardless of which microsatellite dataset was analyzed, or which genetic distance metric was used, no significant IBD was detected (all $P\geq0.130$; Table 6).
*Individual-based metrics.* All instances of significant IBD were limited to *Lynch & Ritland*'s (*1999*) relatedness (note that such relationships are negative). Furthermore, this significant IBD was almost always associated with datasets that included males, with the one exception being the female-only 19-locus high sensitivity dataset (Table 6). Despite relatively strong significance (all $P=0.001$, except for the aforementioned female-only dataset, which had $P=0.031$; Table 6), the nature of relationships between relatedness and geographic distance were consistently weak [augmented dataset with males plus females: slope $=-2\times10^{-6}$, correlation coefficient ($r$) $=0.036$; augmented dataset with males only: slope $=-2\times10^{-6}$, $r=0.037$; low noise dataset with males plus females: slope $=-2\times10^{-6}$, $r=0.020$; low noise dataset with males only: slope $=-2\times10^{-6}$, $r=0.030$; and even weaker for the 19-locus high signal dataset with females only: slope $=-3\times10^{-6}$, $r=0.001$]. Overall, these outcomes are consistent with female-biased dispersal; however, males do not appear to be strongly philopatric.

## DISCUSSION

Spatially explicit population genetic analyses of microsatellite data from invasive insect species can identify the geographic origin(s) and number of independent introductions

**Table 5  Outcomes of genotypic clustering using STRUCTURE (i.e., non-spatial, individual-based), BAPS (i.e., spatial, group- or individual-based) and DAPC (i.e., non-spatial, individual-based).** For STRUCTURE analyses, the number of clusters ($K$) was inferred using both a quantitative (*Evanno, Regnaut & Goudet, 2005*) and a qualitative (*Pritchard & Wen, 2003*) approach. The maximum membership coefficient ($Q$) was used to identify putative "ghost clusters" in cases where the best-fit $K>1$. Means were calculated from either 20 (STRUCTURE) or 10 (BAPS) replicates. For DAPC analyses, abbreviations are Bayesian Information Criterion (BIC) and Principal Components (PCs), and Not Applicable (N/A). Dataset abbreviations are as follows: augmented dataset (ADS; 33 loci), low noise dataset (LNDS; 24 loci), high sensitivity datasets (HSDS; either 10 or 19 loci), males and females (m+f), males only (m), and females only (f).

| Microsatellite dataset | STRUCTURE | | | | | BAPS | | | | DAPC | | | |
| --- | --- | --- | --- | --- | --- | --- | --- | --- | --- | --- | --- | --- | --- |
| | *Evanno, Regnaut & Goudet (2005)* | | *Pritchard & Wen (2003)* | | Maximum $Q$ in each inferred cluster | Groups | | Individuals | | Best-fit $K$ | BIC | No. of PCs retained | No. of clusters per site |
| | Best-fit $K$ | Delta $K$ | Best-fit $K$ | Mean LnP($K$) | | Best-fit $K$ | Mean log(ml) | Best-fit $K$ | Mean log(ml) | | | | |
| ADS m+f | 3 | 10.28 | 3 | −19161.63 | 0.31, 0.33, 0.94 | 1 | −19891.68 | 1 | −19891.68 | 2 | 577.66 | 80 | 2 |
| ADS m | 2 | 48.32 | 2 | −13949.30 | 0.39, 0.92 | 1 | −14533.15 | 1 | −14533.15 | 2 | 419.22 | 120 | 2 |
| ADS f | 2 | 20.73 | 2 | −5204.64 | 0.43, 0.87 | 1 | −5575.48 | 1 | −5575.48 | 1 | 162.61 | N/A | N/A |
| LNDS m+f | 3 | 5.19 | 1 | −11792.72 | 0.45, 0.46, 0.47 | 1 | −12153.79 | 1 | −12153.79 | 3 | 451.90 | 80 | 3 |
| LNDS m | 2 | 87.50 | 2 | −8534.31 | 0.37, 0.93 | 1 | −8861.98 | 1 | −8861.98 | 3 | 329.35 | 40 | 3 |
| LNDS f | 3 | 6.34 | 1 | −3201.78 | 0.38, 0.38, 0.43 | 1 | −3408.28 | 1 | −3408.28 | 1 | 128.88 | N/A | N/A |
| HSDS10 m+f | 3 | 4.98 | 3 | −5345.57 | 0.57, 0.67, 0.98 | 1 | −5549.75 | 1 | −5549.75 | 9 | 242.41 | 45 | 8–9 |
| HSDS10 m | 5 | 2.59 | 5 | −3943.26 | 0.46, 0.47, 0.55, 0.73, 0.86 | 1 | −4086.40 | 1 | −4086.40 | 7 | 183.72 | 35 | 5–7 |
| HSDS10 f | 2 | 0.10 | 1 | −1444.36 | 0.71, 0.80 | 1 | −1529.06 | 1 | −1529.06 | 4 | 71.63 | 20 | 3–4 |
| HSDS19 m+f | 3 | 20.73 | 3 | −10933.57 | 0.35, 0.35, 0.94 | 1 | −11338.45 | 1 | −11338.45 | 4 | 432.41 | 80 | 4 |
| HSDS19 m | 4 | 2.28 | 1 | −7939.17 | 0.37, 0.39, 0.45, 0.89 | 1 | −8242.20 | 1 | −8242.20 | 3 | 314.69 | 60 | 3 |
| HSDS19 f | 2 | 2.01 | 1 | −3035.54 | 0.61, 0.74 | 1 | −3234.04 | 1 | −3234.04 | 2 | 125.00 | 25 | 2 |

**Table 6 Outcomes of Mantel tests of the significance of correlation between geographic and genetic distance.**

| Microsatellite dataset | Isolation-by-distance analysis $P$-values | | | | | |
|---|---|---|---|---|---|---|
| | Population-based metrics | | | Individual-based metrics | | |
| | $D_c$ | $D_s$ | $F_{ST}$[†] | $D_{kf}$ | $D_{ps}$ | $r$ |
| ADS m+f | 0.304 | 0.406 | 0.500 | 0.176 | 0.105 | 0.001[*] |
| ADS m | 0.488 | 0.343 | 0.420 | 0.133 | 0.128 | 0.001[*] |
| ADS f | 0.130 | 0.403 | 0.150 | 0.127 | 0.053 | 0.282 |
| LNDS m+f | 0.259 | 0.271 | 0.356 | 0.414 | 0.342 | 0.001[*] |
| LNDS m | 0.379 | 0.280 | 0.253 | 0.343 | 0.465 | 0.001[*] |
| LNDS f | 0.152 | 0.455 | 0.264 | 0.294 | 0.191 | 0.443 |
| HSDS10 m+f | 0381 | 0.380 | 0.357 | 0.330 | 0.243 | 0.001[*] |
| HSDS10 m | 0.177 | 0.360 | 0.393 | 0.222 | 0.240 | 0.013[*] |
| HSDS10 f | 0.419 | 0.451 | 0.276 | 0.457 | 0.287 | 0.406 |
| HSDS19 m+f | 0.464 | 0.390 | 0.451 | 0.332 | 0.505 | 0.001[*] |
| HSDS19 m | 0.452 | 0.475 | 0.397 | 0.226 | 0.308 | 0.001[*] |
| HSDS19 f | 0.287 | 0.481 | 0.174 | 0.283 | 0.154 | 0.031[*] |

**Notes.**

Abbreviations : ADS; 33 loci, augmented dataset; LNDS; 24 loci, low noise dataset; HSDS; either 10 or 19 loci, high sensitivity datasets; m+f, males and females; m, males only; f, and females only.

[*]Significant $P$-values are marked with an asterisk.

[†]linearized.

into newly invaded areas, and provide insights into the relative importance of natural *vs.* human-mediated dispersal (*e.g.*, fire ants, *Ascunce et al., 2011*; termites, *Perdereau et al., 2013*; hemlock woolly adelgid, *Havill et al., 2016*; *Havill et al. 2021*). These inferences have implications for management, such as enabling targeted control (*e.g.*, of an invasive "bridgehead" population that is the primary source of subsequent spread (*Lombaert et al., 2010*) and which may have evolved higher invasiveness (*Whitney & Gabler, 2008*), or enhanced surveillance of "stowaways" where transportation networks are involved in accelerating spread (*e.g.*, *via* inadvertent movement of contaminated live plants, wood packing materials, or firewood; *Meurisse et al., 2019*). Notably, these insights often depend on the existence of discrete populations, yet a long-standing challenge in population genetics is the "clusters *vs.* clines" problem. This refers to the tendency for clustering methods to fail (or mislead) when structure manifests as continuous gradients of genetic differentiation, and conversely, true signatures of IBD can be distorted if sampling traverses abrupt genetic breaks or is discontinuous and uneven (*e.g.*, *Frantz et al., 2009*; *Bradburd, Coop & Ralph, 2018*). While clusters and clines are not mutually exclusive (see *Rosenberg et al., 2005*), an understanding of the predominant form of spatial-genetic structure informs the choice of appropriate analyses for reconstructing population history. In the case of *D. frontalis*, gaining insights into the species' recent and rapid range expansion to the northeastern United States is of considerable interest owing to the economic and ecological damage this may cause. Accordingly, here we systematically explored alternative explanations for the apparent absence of clusters or clines reported by *Havill et al. (2019)*.

## Clusters

Regardless of whether we increased the number of microsatellite loci (augmented dataset) or analyzed subsets of loci with more favorable signal-to-noise ratio (low noise and high signal datasets), no spatially abrupt genetic subdivisions were detected. Using simulations, *Miller, Cullingham & Peery (2020)* showed that in the presence of even quite low levels of on-going gene flow (*i.e.,* migration rate, $m \geq 0.005$), the best-fit number of clusters inferred using STRUCTURE may be systematically underestimated, leading to erroneous inferences of $K = 1$. Following the recommendations of those authors and others, we compared outcomes across several analytical methods with different assumptions and found that the inferred absence of discrete clusters was robust (Table 5). We interpret this to indicate that either *D. frontalis* does not exhibit discrete population structure, or gene flow is sufficiently high that such structure is not detectable with our data. Other types of molecular markers, such as single nucleotide polymorphisms (SNPs) assayed *via* sub-genomic sampling methods that are applicable to non-model species (*e.g.*, *Elshire et al., 2011*) could reveal subtle fine-scale spatial-genetic structure. For example, compared to inferences from microsatellites, SNPs have identified additional structure in the mountain pine beetle, *D. ponderosae* (*Batista et al., 2016*; *Janes et al., 2018*). That said, although only $4\times$ to $12\times$ more bi-allelic SNPs than multi-allelic microsatellite loci might offer comparable resolution of population structure (*e.g.*, *Liu et al., 2005*; but see *Haasl & Payseur, 2011* for a potentially much higher ratio), microsatellites have desirable properties for assignment tests, kinship analyses and estimating heterozygosity, and therefore remain valuable (*Narum et al., 2008*).

## Clines

When using sampling sites as the unit of analysis, we found that microsatellite dataset composition did not impact conclusions about the absence of gradients of genetic differentiation. This outcome was also robust to the chosen genetic distance metric (Table 6). Range expansion is often considered in the context of a series of founder events that each give rise to a new population with reduced genetic variation along the moving wave of advance, leading to genetic differentiation from the source population. However, when range expansion is recent or on-going, multi-directional, and/or effective population sizes are consistently large, there can be a lag time or weakening of bottleneck effects (also see *Roques et al., 2012* for an example of how Allee effects at a low-density wave front can prevent successive loss of genetic diversity). Likewise, if occasional long-distance dispersal is involved, new populations can establish in leaps and bounds, far from the parent population (*Nichols & Hewitt, 1994*; *Hewitt, 1996*; *Hewitt, 1999*; *Ibrahim, Nichols & Hewitt, 1996*). Under any of these scenarios, IBD is not expected to be immediately detectable. Indeed, the current expansion of *D. frontalis* to the northeastern United States is both recent and on-going: the species spread throughout New Jersey by the mid- to late-2000's, to Long Island in New York by 2014, and was trapped in Connecticut, Rhode Island and Massachusetts by 2015–2016 (*Dodds et al., 2018*). Effective population sizes certainly have the potential to be very large, but little is known about non-outbreak populations, making it difficult to speculate about whether genetic drift would be a strong
driver of allele frequency differences over increasing geographic distances. However, the capacity for long-distance dispersal by *D. frontalis* has been well-documented: this could occur *via* active flight over several kilometers, or *via* passive above-canopy wind-assisted dispersal over tens of kilometers (*Jones et al. 2019*, and references therein). Ultimately, lack of IBD is not entirely unexpected, at least not in the newly invaded portion of this species' current geographic range. However, other explanations exist. For example, detection of IBD might be scale-dependent, whereby its signal fades at increasingly larger geographic distances owing to an upper bound on the maximum attainable genetic distance (*Castric & Bernatchez 2003*). Such non-linearity might be revealed by iterative reanalysis using different distance thresholds (*Van Strien, Holderegger & Van Heck, 2015*), provided that straight-line geographic distance (cf. isolation-by-environment; *Wang & Summers, 2010*) is the primary driver of genetic divergence.

## Female-biased dispersal

Despite overall weak spatial-genetic structure among eastern United States *D. fontalis*, we did find significant individual-based IBD attributable to males (Table 6). Although this inference of female-biased dispersal was limited to *Lynch & Ritland*'s (*1999*) measure of relatedness, it was robust to different compositions of microsatellite loci across datasets. Indeed, this finding is concordant with life history traits of *D. frontalis*. Females are the pioneering sex, who locate a suitable host before recruiting males *via* release of aggregation pheromones (*Jones et al. 2019*). Furthermore, females are larger than males, and given that body mass is correlated with stored energy reserves, their greater capacity for flight (mean distance of 3.4 km *vs.* 2.7 km for males) in a flight mill experiment was attributed to sexual size dimorphism (*Kinn et al., 1994*). Notably, the notion that males are themselves not strongly philopatric is supported by capture-mark-recapture data, which showed no significant differences in dispersal between the sexes over distances up to 1 km (*Turchin & Thoeny, 1993*).

There are several reasons that could explain why individual-based IBD was detectable in our study, yet population-based IBD was not. For instance, this may simply reflect an issue of sample size (*i.e.,* 2,415–32,385 pairwise comparisons among individuals *vs.* only 28–36 among sampling sites depending on the dataset; but note the elevated pseudo-replication associated with the former). Alternatively, a difference in the timescales over which spatial-genetic structure has evolved and/or is assessed may be responsible. Indeed, individual-based genetic distances may better reflect contemporary processes such as intra-generational dispersal, whereas population-based metrics primarily measure historical connectivity given that most signal comes from the accumulated multi-generational effects of dispersal and gene flow (*Sunnucks, 2000*; *Garrick, Caccone & Sunnucks, 2010*; *Epps & Keyghobadi, 2015*). In the context of this study, however, these explanations are plausible only if the other two individual-based genetic distance metrics that we assessed (*i.e.,* $D_{kf}$ and $D_{ps}$) had low power. One potential advantage of *Lynch & Ritland*'s (*1999*) *r* is that loci with more alleles—particularly those with rare alleles—provide more information about relatedness, such that highly polymorphic genetic markers such as microsatellites are very powerful (*Ritland, 2000*). That said, this remains speculative, as simulations have shown

$D_{ps}$ to be more sensitive than $r$ under a variety of IBD scenarios, albeit under simplified conditions (*Shirk, Landguth & Cushman, 2017*).

## Reconciliation with previous work

Our findings for *D. frontalis* are inconsistent with those of *Schrey et al. (2011)* who reported both clusters and clines, in the form of an east–west division approximately coincident with southern Appalachian Mountains and population-based IBD across the eastern United States. Those authors used eight microsatellite loci, all of which were included in *Havill et al.*'s (*2019*) 24-locus dataset, as well as the present study's 33-locus augmented dataset. Generally, trade-offs between the number of loci and number of individuals favor adding loci (*Landguth et al., 2012*), but given the shared loci, it seems unlikely that the marker set alone can account for contrasting inferences. However, geographic and temporal sampling did differ. Whereas *Schrey et al.*'s (*2011*) study design included 19 sites with an average of 63 individuals per site (range: 26–100) collected between fall 2004 to spring 2005, ours included 9 sites with an average of 28 individuals (range: 21–31) collected >10 years later (*i.e.,* summer 2016 to summer 2017, except for Ponte Vedra, Florida, collected in summer 2013). Thus, although the overall spatial scale of sampling was similar between studies (see Introduction), effort allocation was not. Indeed, *Landguth & Schwartz (2014)* have shown that the optimal allocation of individuals for accurate estimation of $F_{ST}$ changes depending on the extent of gene flow limitation, with misallocations leading to either over- or under-estimation of $F_{ST}$, depending on the circumstances. Furthermore, direct comparison between studies may be impacted by the fact that *Schrey et al. (2011)* sampled *D. frontalis* during years of relatively low population densities (albeit following a significant outbreak in 2001–2002) whereas our specimens were collected at a time of considerable increase in trap captures in Louisiana and Alabama, and an active outbreak in Mississippi (B Sullivan, pers. comm., 2020).

During *D. frontalis* outbreak years, pulse irruptions have unusual population dynamics where beetles emerge from infested trees in several waves and inter-tree distance is an important determinant of attack success, with crowded healthy trees being at greater risk owing to the spill-over effects of aggregation pheromones (*Hain et al., 2011*). Conversely, in endemic (non-outbreak) situations, *D. frontalis* persists at low densities and generally must travel further after emergence to encounter trees already under stress (*e.g.,* from lightning strike damage or disease). Indeed, seasonal differences within a given year can also impact dispersal behavior, whereby individuals have the greatest potential for long-distance dispersal in Fall and the lowest potential in mid-Summer, owing to temporal differences in fat content levels (*Turchin & Thoeny, 1993*). Ultimately, the impacts of inter- and intra-year differences in sampling across studies may at least partly account for contrasting conclusions about the nature and strength of spatial-genetic structure.

The idea that preexisting structure may have been "overwritten" either by a change in outbreak status (*i.e.,* from endemic to epidemic) and/or on-going expansion processes underscores the importance of considering both and ecological and geographic context. For example, *James et al. (2015)* showed that while the spruce budworm (*Choristoneura fumiferana*), a cyclical irruptive pest in North America, exhibits weak spatial-genetic

structure, there is considerable nuance to this general finding. For example, the legacy of high connectivity during past outbreaks can persist for some time afterwards, high admixture can be driven by migrant adults sourced from relatively few sites, and different life stages (*i.e.,* adults *vs.* larvae) can concurrently exhibit different degrees of spatial genetic structure. Although the periodicity of spruce budworm outbreaks is considerably longer than that of *D. frontalis* (∼35 years *vs.* ∼5–10 years, respectively) and the synchrony of irruptive populations may occur at different spatial scales, there may be considerable similarity in the underlying processes that generate apparent weak spatial-genetic structure. Future work on *D. frontalis* could attempt to distinguish between migrant and resident individuals, and model range expansion dynamics and associated source–sink dynamics, explicitly incorporating temporal effects.

## CONCLUSIONS

The lack of discrete population differentiation and substantial IBD in *D. frontalis* in the eastern United States has implications for management as the species' impacts spread north. In contrast to alien pests that invade a region *via* long distance introduction by human transport, *D. frontalis* is rapidly spreading into a previously unoccupied region contiguous to its native range. For alien species, gene flow between native and alien populations only occurs *via* additional introductions. Conversely, for *D. frontalis*, the gene flow between long established populations in the southeastern United States and invading populations in New Jersey, New York, and Connecticut is likely to be extensive and on-going. While it is still an open question whether *D. frontalis* had been present in the northeastern areas at low densities but is only now causing impacts due to population increases with less winter mortality, or if it truly represents a recent arrival, the management implications could rely less on looking for differences in beetle genetics, and more on environmental and host tree differences between the south and north. Indeed, this is an approach that seems to be working (*Dodds et al., 2018*).

## ACKNOWLEDGEMENTS

We thank USDA Forest Service for beetle specimens, DeAdra Newman for laboratory assistance, Yale University's DNA Analysis Facility on Science Hill for genotyping services, and Brian Sullivan, Dimitrios Avtzis, Marion Javal, and Patrick James for constructive feedback on the paper.

### Funding

Funding was provided by USDA Forest Service Northern Research Station agreement 18-CR-11242303-089. The funders had no role in study design, data collection and analysis, decision to publish, or preparation of the manuscript.

## Grant Disclosures
The following grant information was disclosed by the authors:
USDA Forest Service Northern Research Station agreement:  18-CR-11242303-089.

## Competing Interests
The authors declare there are no competing interests.

## Author Contributions

- Ryan C. Garrick conceived and designed the experiments, analyzed the data, prepared figures and/or tables, authored or reviewed drafts of the paper, and approved the final draft.
- Ísis C. Arantes performed the experiments, analyzed the data, authored or reviewed drafts of the paper, and approved the final draft.
- Megan B. Stubbs performed the experiments, authored or reviewed drafts of the paper, and approved the final draft.
- Nathan P. Havill conceived and designed the experiments, authored or reviewed drafts of the paper, and approved the final draft.

## Field Study Permissions
The following information was supplied relating to field study approvals (i.e., approving body and any reference numbers):

All beetle samples were collected with permission from the U.S. Department of Agriculture Forest Service. Trapping was done as part of the U.S. Forest Service Southern Pine Beetle Prevention and Restoration Program (see: http://southernforesthealth. net/insects/southern-pine-beetle/southern-pine-beetle-prevention-and-restoration-program). Each of the collectors were employees of the entities from which the samples were collected, so therefore had permission to collect at those sites:

Site; Collector

Sicily Island (U.S. National Forest); Jim Meeker, USDA Forest Service

Homochitto (U.S. National Forest); JoAnne Barrett, USDA Forest Service

Holly Springs (U.S. National Forest); Jim Meeker, USDA Forest Service

Tombigbee (U.S. National Forest); Jim Meeker, USDA Forest Service

Talladega (U.S. National Forest); Larry Spivey, USDA Forest Service

Woolford (Maryland Little Choptank River Sactuary); Heather Disque, Maryland Department of Agriculture)

Goat Hill (Pennsylvania Barrens Preserve); Gina Peters and Paul Smith (Pennsylvania Department of Environmental Protection)

Warwick (U.S. National Forest); Chip Bates, USDA Forest Service

Ponte Vedra (Florida State Land); Jiri Hulcr, University of Florida

## Data Availability
All southern pine beetle (D. frontalis) specimen and geographic sampling site information, and nuclear microsatellite genotypes, are available at Dryad: Garrick, Ryan;

Arantes, Ísis; Stubbs, Megan; Havill, Nathan (2021), Weak spatial-genetic structure in a native invasive, the southern pine beetle (Dendroctonus frontalis), across the eastern United States, Dryad, Dataset, https://doi.org/10.5061/dryad.tqjq2bvxx.
    Previously published DNA sequences are available at NCBI BioProject: PRJNA493650.

## Supplemental Information
Supplemental information for this article can be found online at http://dx.doi.org/10.7717/peerj.11947#supplemental-information.

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
