# Peer review of "Weak spatial-genetic structure in a native invasive, the southern pine beetle (Dendroctonus frontalis), across the eastern United States"

_PeerJ, doi:10.7717/peerj.11947_

## Round 0.1 · original submission · Major Revisions

Dear Dr. Garrick and colleagues:

Thanks for submitting your manuscript to PeerJ. I have now received three independent reviews of your work, and as you will see, the reviewers raised some concerns about the research (and manuscript). Despite this, these reviewers are optimistic about your work and the potential impact it will have on research studying southern pine beetle ecology and populations genetics. Thus, I encourage you to revise your manuscript, accordingly, taking into account all of the concerns raised by all three reviewers.

There seems to mainly be missing comparisons with other relevant studies. These are listed by the reviewers and should be acknowledged and included in your analyses/observations/conclusions. Also, more description of the species is needed, as well as your methodology. Please also expand your observations of the data within a broader ecological context.

Therefore, I am recommending that you revise your manuscript, accordingly, taking into account all of the issues raised by the reviewers.

Good luck with your revision,

-joe

·

Basic reporting

The manuscript of Garrick et al. provides a very interesting and comprehensive analysis on the populations structure of Dendroctonus frontalis across the eastern United States. The language used describes efficiently the concept, methods and results of this analysis, whereas the authors cite all the necessary lliterature references. I believe however, that the introductory part should be somehow complemented with a few more information regarding D. frontalis itself, before explaining all the previous studies that were done on this species.

Experimental design

The experimental design is really robust. The authors address each approach individually, and explain in details what they will do and why they chose this approach. This works excellently in their case, as they leave nothing vague or questionable, something that facilitates greatly the comprehension of their work. This is further enhanced by providing the settings employed in each software, something that allows the replicate of their analysis. Nevertheless, what I would really like to see included in the Materials and Methods part, are a couple of sentences describing the process used when sampling the D. frontalis individuals. In my opinion, this is important particularly in phylogeographic studies, and should be added, .

Validity of the findings

Following the structure they adopted in the description of the Experimental Design, the authors clearly address each approach, and explain their findings. It is also very important, that their speculation regarding the detection of IBD when only females were included, is learly identified as such, leaving no doubt about that.

Additional comments

The manuscript of Garrick and colleagues provides a very interesting and comprehensive analysis of the population structure of D. frontalis in the eastern United States. The authors adopt a very convenient and reader-friendly format to present the methods used, presenting each approach individually, providing the necessary information therein. This becomes even more important in the next parts (Results and Discussion) as by doing that, they facilitate reader to easily and effectively understand how the authors have reached to these conclusions. The only thing I would like to the authors to add, is some more information regarding D. frontalis in the Introduction, but also some details about the sampling protocol of the individuals analysed.

·

Basic reporting

The article by Garrick et al. assessed several explanations for the absence of spatial genetic structure in a native pest, Dendroctonus frontalis.
The article is well written and includes enough background to demonstrate how the work is relevant for the field and for the management of D. frontalis.

Experimental design

The methods and experimental design are clearly explained. There are many stages of analysis but the structure of the text is very clear. The choice of methods is well justified.

Validity of the findings

The manuscript confirms the lack of discrete population differentiation and isolation by distance in D. frontalis, based on several, complementary methods, and the authors give several explanations for this result.
As indicated by the authors, these results have strong implications for the management of D. frontalis

Additional comments

Perhaps a paragraph should be added in the introduction on the biology of the species, with a focus on the traits that may affect its genetic structure. The authors already mentionned dispersal distances l59-61, but other traits such as temperature tolerance or reproductive mode could also have an influence.

L404-417 - identifying the best-fit value of K : read Janes et al. (2017) and Tsykun et al. (2019) for an illustration of the method.
--> Janes, J. K. et al. The K = 2 conundrum. Mol Ecol 26, 3594–3602, https://doi.org/10.1111/mec.14187 (2017).
--> Tsykun, T. et al. Fine-scale invasion genetics ofthe quarantine pest, Anoplophora glabripennis, reconstructed in single outbreaks. Scientific reports. 9:19436 (2019)

L462-463 – Add a reference like Lombaert et al. (2010) for bridgehead effect, and like Whitney & Gabler (2008) for rapid evolution in introduced species.
--> Lombaert, E. et al. Bridgehead effect in the worldwide invasion of the biocontrol harlequin ladybird. PLoS One 5:e9743 (2010)
--> Whitney, KD & Gabler, CA Rapid evolution in introduced species, “invasive traits” and recipient communities: Challenges for predicting invasive potential. Divers. Distrib. 14:569-580 (2008)

L508 – see also Roques et al (2012)
--> Roques, L. et al. Allee effect promotes diversity in traveling waves of colonization. PNAS 109 :8828-8833 (2012)

L571-572 – Remove one « Generally »

·

Basic reporting

no comment - see below

Experimental design

no comment - see below

Validity of the findings

no comment - see below

Additional comments

The authors provide a detailed analysis of the population genetic structure of the southern pine beetle in the eastern US with the goal of reconciling different results found in the same system. The manuscript is well written and well-organized.

I have two main concerns about this manuscript.

First, I sincerely appreciate the attention to detail and the well-organized approach that was taken in this manuscript. However, I do have concern that these detailed investigations were not undertaken in the previous analysis and associated publications. The onus should be on an author to explore these types of questions when first reporting on the phenomenon. As written, this is a (well-executed) technical piece of work, however, without much of a conceptual contribution. For that reason, while the science undertaken is acceptable, I question the strategy that is being taken here.

Second, related to conceptual contributions, I was a bit surprised that that main line of inquiry was related to techno/analytical solutions to this problem. The assumption seems that if two different studies found different patterns of spatial genetic structure, then something must be wrong with our molecular markers. Ecological context (i.e., timing and outbreak status) was surprisingly not considered. As indicated above, while I do appreciate the due diligence of verifying that the markers and analytical approach were not at fault, I was surprised that more attention wasn’t given to alternate ecological and demographic explanations.

One very logical explanation from my perspective is that the two studies that precipitated this analysis were conducted at different points in time during and outbreak (i.e., 2011 vs. 2019). This idea is mentioned near the end of the MS (line 580 +). Given that you eliminated all other possible explanations, I feel that this explanation should be given greater attention. The spatial genetic consequences of cyclic and irruptive populations are not yet well understood, but I feel that you have likely landed on the most plausible answer in your comment about existing structure being “overwritten” (line 584). Your work may not be “complicated” by this issue of different timings of sampling, but in fact may be driven by them. Please see James et al (2015) and Larroque et al (2019) in this regard. I encourage you to explore this idea in more detail to not only better describe the interesting differences in SGS in D. frontalis at different points of time, but also to add further conceptual content and rigour to this work.

James PMA, Cooke B, Brunet B, Lumley L, Sperling FAH, Fortin M-J, Quinn V, Sturtevant BR. 2015. Life-stage differences in spatial genetic structure in an irruptive forest insect: Implications for dispersal and spatial synchrony. Molecular Ecology. 24(2): 296-309.

Larroque J, Legault S, Johns R, Lumley L, Cusson M, Renaut S, James PMA. 2019. Temporal variation in spatial genetic structure during population outbreaks: distinguishing among different potential drivers of spatial synchrony. Evolutionary Applications 12(10): 1931-1945

---

## Round 0.2 · Minor Revisions

Dear Dr. Garrick and colleagues:

Thanks for revising your manuscript. Two reviewers are very satisfied with your revision (as am I). Great! However, there are some concerns raised by the third reviewer (a few minor by reviewer 2 as well). Please address these and submit a revision ASAP.

Best,

Good luck with your revision,

-joe

·

Basic reporting

The revised version of the manuscript exhibits no further points that need to be improved or enhanced, as the authors have done a very nice work addressing the points raised previsouly. The manuscript is written in a very clear and unambiguous manner, and the literature references are as required. the

Experimental design

The initial concept/question behind this work is something logical that needs to be resolved, and this is very clearly explained in Introduction (Lines 81-93).
The experimental design behind the analysis of this manuscript is very meticulously designed and performed. In particular, the authors pay much attention to the handling of microsatellites in order to gain the most of them, avoiding any possible flaw might have reduced the clarity of the conclusions.

Validity of the findings

As the use of microsatellites might sometimes produce misleading results, the authors have carefully employed numerous methods and protocols that they use in their analysis to avoid such kind of pitfalls (e.g. Lines 255-256/ Lines 267-278). These additional add-ons significantly support the validity of their findings.

Additional comments

Overall, the manuscript by Garrick and colleagues provides a very thorough and interesting investigation on the intra-population structure of D. frontalis across the eastern US, based on the up-to-date approaches in phylogeographic and phylogenetic analyses. For that, I suggest that the manuscript can now be accepted for publication.

·

Basic reporting

The article is very well written, and gives key information for managing a major insect pest.
I particularly appreciated the presentation of the methods, and the discussion of the results, especially the comparison with previous studies.
The authors have responded to all the comments in the previous review.
I therefore recommend this article for publication.
I have just three minor points to note:
l95-109: this paragraph might have been better placed in the discussion, in the "reconciliation with previous work" section for instance. It would make the introduction shorter, and easier to read. This is only a suggestion, however.
l153-156: it is not clear who sampled the specimens and when.
l224: check grammar

Experimental design

no comment

Validity of the findings

no comment

·

Basic reporting

see below

Experimental design

see below

Validity of the findings

see below

Additional comments

While some of my previous comments were taken into consideration, my impression is that they were done so only superficially. The focus remains highly technical in nature and still does not adequately address the questions at hand. Greater clarity and an improved framing of the study is needed. Doing the same analysis with just more markers seems like a poor justification to undertake this work, especially when the value of these markers is put into question given the framing regarding previous contradictory results. Similarly, highlighting the contribution of this re-examination as a “roadmap” (line 140) seems strange, as eliminating potential alternate explanations for observed results is just good science.

Specific comments
Line 44 – and outbreak dynamics?
Line 53 – suggest adding your research objective/question here (e.g., statement on line 72) and starting a new paragraph to present life history information.
Line 75 – development “and application” of molecular tools?
Line 100 – what is a “biologically meaningful distance estimate” and how does one distinguish from an un-meaningful one?
Line 107 –context and citation needed for reference to rapid evolution
Line 113 – Here you refer to the unexpected absences as the motivating problem, but is there any a priori reason to consider the absence of structure to be more likely than the presence of structure? One could also frame this study to examine the unexpected structure identified in the first study. Is it not plausible that technical errors could have resulted in spurious detection of SGS? Why is the absence of structure so much more suspect than its presence?
Line 121 – “albeit ephemerally” – do you know this for sure? Suggest remove
Line 122 – reference needed for this idea
Line 126 – why “notwithstanding ... panmixia”? You refer to panmixia above as a potential alternate hypothesis. You could have both panmixia and weak markers, no?
Line 131 – “only those loci …” – Precedent for this approach? Reference needed
Line 140 – seems strange to highlight the value of new molecular markers. Suggest re-emphasize conceptual (vs technical) contributions.
Line 185 – the absence of consideration of outbreak dynamics and demography as an explanation for the differences between the two studies suggests to me that my previous comments were only superficially taken into consideration. Although referred to int eh discussion, this concept should be woven into the manuscript more thoroughly. Population connectivity and effective population size vary through time in irruptive species, and both of these factors affect population genetic inference. The focus on uSat number and quality misses the mark and does little to actually address your question.

---

## Round 0.3 · accepted · Accept

Dear Dr. Garrick and colleagues:

Thanks for revising your manuscript based on the concerns that were raised. I now believe that your manuscript is suitable for publication. Congratulations! I look forward to seeing this work in print, and I anticipate it being an important resource for groups studying southern pine beetle ecology and populations genetics. Thanks again for choosing PeerJ to publish such important work.

Best,

-joe

---

## Author Rebuttal · Round 0.3

THE UNIVERSITY of
**MISSISSIPPI**
DEPARTMENT OF BIOLOGY

Dr. Joe Gillespie
Academic Editor, *PeerJ*
Department of Microbiology and Immunology
University of Maryland School of Medicine
Baltimore, Maryland, USA

**Re: revision of *PeerJ* manuscript # 2021:01:57327:2 (Garrick et al.)**

Dear Dr. Gillespie,

Thank you for the opportunity to submit a second revision of our paper. Here we provide a point-by-point description of how reviewers' comments have been addressed. In some instances, we have a different point of view to one of the reviewers regarding how our paper should have been framed, and what is important or worth highlighting. Accordingly, we give some more rationale for why we did what we did. As before, a version of the main text showing all edits as "track changes" has been included. Thank you handling this paper.

Sincerely,

Ryan Garrick, for the authors

**Academic editor's comments**

*>> Two reviewers are very satisfied with your revision (as am I). Great! However, there are some concerns raised by the third reviewer (a few minor by reviewer 2 as well). Please address these and submit a revision ASAP.*

Thank you. We have made all suggested minor edits, and made further attempts to clarify our position with respect to lingering concerns of Rev 3, as described below.

**Reviewers' comments**

**Reviewer #1 (Avtzis)**

*>> Overall, the manuscript by Garrick and colleagues provides a very thorough and interesting investigation on the intra-population structure of D. frontalis across the eastern US, based on the up-to-date approaches in phylogeographic and phylogenetic analyses. For that, I suggest that the manuscript can now be accepted for publication.*

Thank you for the earlier feedback.

**Reviewer #2 (Javal)**

*>> The authors have responded to all the comments in the previous review. I therefore recommend this article for publication. I have just three minor points to note:*
*l95-109: this paragraph might have been better placed in the discussion, in the "reconciliation with previous work" section for instance. It would make the introduction shorter, and easier to read. This is only a suggestion, however.*

We appreciate this suggestion, but we have opted to leave this information in the Introduction, as we feel that it gives the reader important background information that is directly relevant to our stated goal of this study (see Abstract). Briefly, we wanted to systematically assess several plausible alternative explanations for apparent conflict between two previous studies re: the existence vs. absence of spatial-genetic structure in the southern pine beetle. This section of text provides details on the near equivalency between those studies in terms of some of the most obvious potential reasons for the discrepant findings (i.e., underlying genetic marker set; extent of geographic sampling; sample size adequacy). We think that stating this early, in the Introduction, provides key rationale for focusing on other more nuanced explanations (i.e., information content of loci, null alleles, homoplasy, etc.). This section of text also provides rationale as to why we should care about the discrepant outcomes in terms of the different types of on-the-ground management interventions that would be appropriate under each scenario.

*>> l153-156: it is not clear who sampled the specimens and when.*

Coinciding with submission of the first revised version of this paper, the *PeerJ* journal office asked us for clarification on a similar topic> Based on our response, they have now added the following text into the field permit statement:

*"All beetle samples were collected with permission from the U.S. Department of Agriculture Forest Service. Trapping was done as part of the U.S. Forest Service Southern Pine Beetle Prevention and Restoration Program (see: [http://southernforesthealth.net/insects/southern-pine-beetle/southern-pine-beetle-prevention-and-restoration-program](http://southernforesthealth.net/insects/southern-pine-beetle/southern-pine-beetle-prevention-and-restoration-program)). Each of the collectors were employees of the entities from which the samples were collected, so therefore had permission to collect at those sites: Site; Collector Sicily Island (U.S. National Forest); Jim Meeker, USDA Forest Service Homochitto (U.S. National Forest); JoAnne Barrett, USDA Forest Service Holly Springs (U.S. National Forest); Jim Meeker, USDA Forest Service Tombigbee (U.S. National Forest); Jim Meeker, USDA Forest Service Talladega (U.S. National Forest); Larry Spivey, USDA Forest Service Woolford (Maryland Little Choptank River Sactuary); Heather Disque, Maryland Department of Agriculture) Goat Hill (Pennsylvania Barrens Preserve); Gina Peters and Paul Smith (Pennsylvania Department of Environmental Protection) Warwick (U.S. National Forest); Chip Bates, USDA Forest Service Ponte Vedra (Florida State Land); Jiri Hulcr, University of Florida"*

We have also edited the main text in this latest revision to point the reader to this statement, and clarified the collection dates. Taken together, we think this has now addressed the reviewer's concern.

*>> l224: check grammar*

Thank you. We have edited that sentence to improve readability.

**Reviewer #3 (James)**

*>> While some of my previous comments were taken into consideration, my impression is that they were done so only superficially. The focus remains highly technical in nature and still does not adequately address the questions at hand. Greater clarity and an improved framing of the study is needed. Doing the same analysis with just more markers seems like a poor justification to undertake this work, especially when the value of these markers is put into question given the framing regarding previous contradictory results. Similarly, highlighting the contribution of this re-examination as a "roadmap" (line 140) seems strange, as eliminating potential alternate explanations for observed results is just good science.*

Our research questions were indeed technical in nature. That was our goal and it has not changed, so we do not agree with the premise here. In general, we are not in favor of retroactively rewriting the research questions, as that seems somewhat disingenuous. In our opinion, the rationale and justification for the framing of the paper were sound, and so we would prefer to remain true to our original goals.

To suggest that we merely did the same analyses as in previously published papers, just with more makers, is an inaccurate characterization of this work. As we have stated before, we 1) **expanded the set of analyses** (e.g., DAPC clustering, individual-based IBD were not previously done), 2) **evaluated novel partitions of the data** (e.g., male-only, female-only sets were not previously analyzed), and 3) **systematically assessed potential causes of failure to detect structure inherent to the markers themselves** (i.e., null alleles, allele size homoplasy). Our intention was always to present a thorough study that focused on understanding the extent to which technical artefacts (signal to noise considerations, etc.) could reconciling discrepant results from previous work. We certainly agreed with the suggestion to add value to the paper via a more detailed discussion of ecological context (timing and outbreak status), and so we did that in the previous revision, at what we felt was appropriate depth (more on this below).

We agree that our reference to a "roadmap" was an overstatement, and we have now removed that text.

*>> Line 44 – and outbreak dynamics?*

We have now edited the text to include this as well.

*>> Line 53 – suggest adding your research objective/question here (e.g., statement on line 72) and starting a new paragraph to present life history information.*

We have now reorganized and split this paragraph as suggested.

*>> Line 75 – development "and application" of molecular tools?*

We have now edited the sentence (which was shifted up) to include application as well.

*>> Line 100 – what is a "biologically meaningful distance estimate" and how does one distinguish from an un-meaningful one?*

It is one that is based on adequate sample sizes such that reasonable precision and accuracy can be expected (cf. an estimated value that is quite meaningless owing to small sample size effects).

Given our phrasing here ("…*in both studies, per-site sample sizes were quite large… Thus, it would seem that biologically meaningful genetic distance estimates should have been attainable*"), we feel that there is no strong ambiguity here, and therefore chose to leave it as-is.

*>> Line 107 –context and citation needed for reference to rapid evolution*

Agreed. We now specify rapid evolution in novel environments leading to larger population sizes and/or faster expansion speeds, and provided a representative citation for this "evolution first" scenario (i.e., Szücs et al. 2017 PNAS 114:13501).

*>> Line 113 – Here you refer to the unexpected absences as the motivating problem, but is there any a priori reason to consider the absence of structure to be more likely than the presence of structure? One could also frame this study to examine the unexpected structure identified in the first study. Is it not plausible that technical errors could have resulted in spurious detection of SGS? Why is the absence of structure so much more suspect than its presence?*

This is a good question. Indeed, in the phylogeographic literature, a broadly distributed panmictic population is often invoked as the "starting condition" for subsequent vicariance events that lead to lineage splitting / emergence of finer-scale structure. One problem we see with this notion (and the reason for our inclination to expect spatial genetic structure in SPB *a priori*) is that there are very few empirical examples of species that truly have a widely distributed panmictic population (i.e., spanning a broad range of latitudes and longitudes, and the inherent environmental heterogeneity that this encompasses). While the American eel and European eel (Côté et al. 2013 Mol Ecol 22:1763) are exceptions to this, these species have unusual traits (e.g., catadromous life cycles) that play are large role in maintaining this rare widespread panmixia. Even for highly mobile taxa such as microbes, isolation-by-distance (e.g., among OTUs within a species) is common. Accordingly, we think the most pertinent question is less about "is there IBD" and more about "over what spatial scale is IBD detectable"? We consider this, as well the border version of it that also applies to clusters (i.e., *how* is genetic variation spatially structured, cf. *is* genetic variation spatially structured?) to be a useful *a priori* expectation for our work on SPB. For this reason, we focused on "…explanations for an unexpected absence of clusters or clines…".

*>> Line 121 – "albeit ephemerally" – do you know this for sure? Suggest remove*

We have tried to strike a balance here, between wanting to highlight the possibility of this re-partitioning of genetic variation being transient vs. completely absent over time, by replacing with "perhaps ephemerally".

*>> Line 122 – reference needed for this idea*

We have now added citations that draw a comparison with impacts of postglacial expansion into newly available habitats (Hewitt 1996 Biol J Linn Soc 58:247, and Hewitt 2004 Philos Trans R Soc Lond B Biol Sci 359, 183). For completeness, we also noted a counter-example where range expansion *generates* spatial-genetic structure, rather than overwrites it (Excoffier & Ray 2008 TREE 23: 347).

*>> Line 126 – why "notwithstanding ... panmixia"? You refer to panmixia above as a potential alternate hypothesis. You could have both panmixia and weak markers, no?*

Thank you for pointing this out, it was a poor choice of word that obscured our intended meaning. We have edited this to now read "… and acknowledging the potential for genuine panmixia (i.e., a null hypothesis that may be true)…". We think this makes it clearer that we recognize the real possibility of that a lack detectable

spatial-genetic structure my simply be because there really is none, and that we use this latter scenario as the null hypothesis against which to test alternative explanations.

>> *Line 131 – "only those loci …" – Precedent for this approach? Reference needed*

In the previous versions of this paper, we had provided a reference for this approach (i.e., Russello et al. 2012 Evol Appl 5:39) in the Materials and Methods (L267), but we see that it might be too late there, and so we have added the same at that citation to an earlier location, here in the Introduction, as suggested.

>> *Line 140 – seems strange to highlight the value of new molecular markers. Suggest re-emphasize conceptual (vs technical) contributions.*

In this instance we disagree with the implication that new, carefully validated and characterized molecular markers are of little value. While this may simply be a difference in personal option, we do not consider out take on this to be unusual. Accordingly, we have chosen to retain our original text (except for removing an earlier reference to a "roadmap", as mentioned above).

>> *Line 185 – the absence of consideration of outbreak dynamics and demography as an explanation for the differences between the two studies suggests to me that my previous comments were only superficially taken into consideration. Although referred to int eh discussion, this concept should be woven into the manuscript more thoroughly. Population connectivity and effective population size vary through time in irruptive species, and both of these factors affect population genetic inference. The focus on uSat number and quality misses the mark and does little to actually address your question.*

We feel that our question was pretty clearly stated, technical in nature, and that we directly address it. This criticism seems to be based on the suggestion that we had a different question at the outset (or that we should now change it retroactively), which is incorrect. Given this, we agree that we have missed the mark on intended research question. We believe it reasonable for us to insist that we stay true to the original goals of this paper. The previous revision made a genuine, substantive, attempt to integrate deeper *discussion* of ecological context. Specifically, in our previous response letter, we wrote the following:

*"We agree that outbreak status, and the potential for this to have differed across studies at the time of sampling, warrants an expanded treatment. We have added details to the Discussion that more thoroughly explains the consequences of endemic vs. epidemic conditions on dispersal, and we also included a brief comment on seasonal (i.e., intra-year) changes in population dynamics that might also have contributed to apparent differences among studies."*

*"… we certainly see a clear connection between our study species and the concepts that apply to the spruce budworm system. Accordingly, we have added a paragraph to the Discussion that highlights the importance of geographic and ecological context, and also flags the more nuanced view of weak spatial genetic structure in cyclical irruptive insect pests…"*

Each of these were addressed via stand-alone newly added paragraphs with relevant citations. Together, these constitute ~17% of the entire Discussion section. Accordingly, rather than being superficial, we think this was done at an appropriate depth.